# Distinct Metabolic Profile Associated with a Fatal Outcome in COVID-19 Patients during the Early Epidemic in Italy

Elisa Saccon,[a] Alessandra Bandera,[b,c,d] Mariarita Sciumè,[e] Flora Mikaeloff,[a] Abid A. Lashari,[f] Stefano Aliberti,[c,g] Michael C. Sachs,[f] Filippo Billi,[h] Francesco Blasi,[c,g] Erin E. Gabriel,[f] Giorgio Costantino,[i,j] Pasquale De Roberto,[e] Shuba Krishnan,[a] Andrea Gori,[b,c,d] Flora Peyvandi,[c,k] Luigia Scudeller,[l] Ciro Canetta,[h] Christian L. Lorson,[m,n] Luca Valenti,[c,o] Kamal Singh,[m,n] Luca Baldini,[e,p] Nicola Stefano Fracchiolla,[e] on behalf of the COVID-19 Network Working Group, Ujjwal Neogi[a,m]

[a]Division of Clinical Microbiology, Department of Laboratory Medicine, Karolinska Institute, Stockholm, Sweden
[b]Infectious Diseases Unit, Fondazione IRCCS Ca' Granda Ospedale Maggiore Policlinico, Milan, Italy
[c]Department of Pathophysiology and Transplantation, University of Milan, Milan, Italy
[d]Centre for Multidisciplinary Research in Health Science (MACH), University of Milan, Milan, Italy
[e]Hematology Unit, Fondazione IRCCS Ca' Granda Ospedale Maggiore Policlinico, Milan, Italy
[f]Department of Medical Epidemiology and Biostatistics, Karolinska Institutet, Stockholm, Sweden
[g]Internal Medicine Department, Respiratory Unit and Cystic Fibrosis Adult Center, Fondazione IRCCS Ca' Granda Ospedale Maggiore Policlinico, Milan, Italy
[h]Acute Medical Unit, Department of Medicine, Fondazione IRCCS Ca' Granda Ospedale Maggiore Policlinico, Milan, Italy
[i]Department of Anesthesia, Critical Care and Emergency, Fondazione IRCCS Ca' Granda Ospedale Maggiore Policlinico, Milan, Italy
[j]Department of Clinical Sciences and Community Health, University of Milan, Milan, Italy
[k]Angelo Bianchi Bonomi Hemophilia and Thrombosis Center, Fondazione IRCCS Ca' Granda Ospedale Maggiore Policlinico, Milan, Italy
[l]Clinical Trials Team, Scientific Direction, Fondazione IRCCS Ca' Granda Ospedale Maggiore Policlinico, Milan, Italy
[m]Bond Life Sciences Center, University of Missouri, Columbia, Missouri, USA
[n]Department of Veterinary Pathobiology, University of Missouri, Columbia, Missouri, USA
[o]Precision Medicine Unit, Department of Transfusion Medicine and Hematology, Fondazione IRCCS Ca' Granda Ospedale Maggiore Policlinico, Milan, Italy
[p]Department of Oncology and Hemato-oncology, University of Milan, Milan, Italy

Nicola Stefano Fracchiolla and Ujjwal Neogi contributed equally to this article. Author order was determined alphabetically.

**ABSTRACT** In one year of the coronavirus disease 2019 (COVID-19) pandemic, many studies have described the different metabolic changes occurring in COVID-19 patients, linking these alterations to the disease severity. However, a complete metabolic signature of the most severe cases, especially those with a fatal outcome, is still missing. Our study retrospectively analyzes the metabolome profiles of 75 COVID-19 patients with moderate and severe symptoms admitted to Fondazione IRCCS Ca' Granda Ospedale Maggiore Policlinico (Lombardy Region, Italy) following SARS-CoV-2 infection between March and April 2020. Italy was the first Western country to experience COVID-19, and the Lombardy Region was the epicenter of the Italian COVID-19 pandemic. This cohort shows a higher mortality rate compared to others; therefore, it represents a unique opportunity to investigate the underlying metabolic profiles of the first COVID-19 patients in Italy and to identify the potential biomarkers related to the disease prognosis and fatal outcome.

**IMPORTANCE** Understanding the metabolic alterations occurring during an infection is a key element for identifying potential indicators of the disease prognosis, which are fundamental for developing efficient diagnostic tools and offering the best therapeutic treatment to the patient. Here, exploiting high-throughput metabolomics data, we identified the first metabolic profile associated with a fatal outcome, not correlated with preexisting clinical conditions or the oxygen demand at the moment of diagnosis. Overall, our results contribute to a better understanding of COVID-19-related metabolic disruption and may represent a useful starting point for the identification of independent prognostic factors to be employed in therapeutic practice.

Address correspondence to Nicola Stefano Fracchiolla, nicola.fracchiolla@policlinico.mi.it, or Ujjwal Neogi, ujjwal.neogi@ki.se.

SystemsVirologyLab @NeogiLab

**KEYWORDS** COVID-19, Italy, metabolomics, fatal outcome, predictive biomarkers

Coronavirus disease 2019 (COVID-19), caused by infection with the novel severe acute respiratory syndrome coronavirus 2 (SARS-CoV-2), caused a massive worldwide pandemic after its identification in early 2020. Although most individuals remain asymptomatic or display mild symptoms, 15 to 20% of patients exhibit severe symptoms, specifically respiratory distress, often requiring mechanical ventilation and/or intensive care unit (ICU) admission (1), with a mortality rate after ICU admission estimated around 40% (2). Multiple studies have identified profound underlying conditions that demonstrate increased susceptibility to a more severe prognosis and a higher risk of fatality, including the male gender, old age (3, 4), and certain underlying medical conditions such as hypertension, cardiovascular diseases, diabetes, and obesity (4, 5). Additionally, patients infected with SARS-CoV-2 present metabolic dysregulation, possibly due to immune-triggered inflammation or other changes in the host physiology, and these alterations often reflect the disease severity (1, 6–8). For instance, levels of particular amino acids positively correlated with severe COVID-19 cases (1, 9, 10). Moreover, perturbations in energy metabolisms, tricarboxylic acid (TCA) and urea cycles (8, 11), and lipid metabolites are also correlated with the disease prognosis (1, 9, 12). Thus, it is essential to assemble a complete metabolic signature correlated with disease severity to identify a set of biomarkers strongly associated with the patient outcome, with the final goal of employing them for diagnostic and therapeutic purposes.

In this study, we performed untargeted plasma metabolomics using global metabolomics data (Discovery HD4; Metabolon, USA) collected from 75 COVID-19 patients during their admission to Fondazione IRCCS Ca' Granda Ospedale Maggiore Policlinico (Italy) from March to April 2020 (ethical clearance no. 462-2020-bis), as previously described (13). SARS-CoV-2 positivity was confirmed by PCR. Clinical data were collected from the registries. Logistic regression was used to model associations of each biomarker with COVID-19-related in-hospital mortality, adjusting for age, gender, and body mass index (BMI). One biomarker was included in each model, but all other covariates were prespecified for inclusion based on previous literature suggesting that they are potential confounders. As BMI was missing for many patients, multiple imputation by chained equations (MICE) was used (via the MICE R package) to impute BMI, using the predictive mean matching method. Imputation was done in two stages, first including no biomarkers and then including those biomarkers that were found to be significant under the original imputation (see Table 1 for clinical and demographic data). Fifty imputed data sets were created, and the resulting pooled estimates and inferences were combined using Rubin's rules. $P$ values were not adjusted for multiple comparisons in this analysis.

Metabolomics data were $\log_2$ transformed and plotted using histograms with normal distribution superimposed. The R package LIMMA was applied for differential abundance analysis between different mask types (nasal cannula/VentMask/continuous positive airway pressure [CPAP]), outcome (survivors/nonsurvivors), and severity (moderate/severe). Adjustment for multiple testing was assessed using a false discovery rate (FDR) of <0.05. A heat map was built using the R package ComplexHeatmap. Uniform Manifold Approximation and Projection (UMAP) representations were assembled using the R package UMAP.

Metabolites with a variance equal to zero were removed, and positive significant pairwise correlations after Bonferroni correction (Spearman, adjusted $P < 0.00001$) were used for association analysis. The strength of the connections was evaluated by plotting the distribution of the correlation coefficients in a graphical network using igraph (https://igraph.org/python/). The network was compared to a random network with similar dimensions to validate that the structure of the network was not due to chance. Community detection was performed using the Leiden algorithm (https://leidenalg.readthedocs.io/en/stable/index.html). For each community large enough ($n > 30$), metabolite set enrichment analysis (MSEA) with KEGG and Metabolon terms via the

**TABLE 1** Clinical and demographic parameters of the study populations

| Characteristic[a] | Survivors | Nonsurvivors | P value[b] |
|---|---|---|---|
| No. of participants | 57 | 18 | |
| Gender (%) | | | 0.1169 |
| Male | 17 (30.4) | 9 (50) | |
| Female | 40 (69.4) | 9 (50) | |
| Age in yrs (median [IQR]) | 62 (52–73.5) | 76.5 (68.5–82) | 0.0013 |
| BMI, median (median [IQR])[c] | 26.3 (24.4–29.5) | 29.6 (27–32.2) | 0.1697 |
| SpO$_2$, median (median [IQR])[c] | 97 (94.25–98) | 95 (89.25–98) | 0.0768 |
| pH, median (median [IQR])[c] | 7.48 (7.45–7.5) | 7.49 (7.45–7.53) | 0.6672 |
| Lactate, median (median [IQR])[c] | 0.9 (0.7–1.1) | 1.5 (1.1–1.8) | <0.0001 |
| Comorbidities (%) | 38 (66.7) | 17 (94) | 0.0297 |
| Hypertension | 26 (46.4) | 9 (50) | 0.7913 |
| Diabetes | 9 (16) | 3 (16.7) | NS |
| Lung disease | 4 (7.1) | 1 (5.6) | NS |
| COPD[d] | 4 (7.1) | 2 (11.1) | 0.6257 |
| Obesity | 8 (14.3) | 6 (33) | 0.0867 |
| Renal disease | 4 (7.1) | 3 (16.7) | 0.3481 |
| Liver disease | 1 (1.8) | 1 (5.6) | 0.4249 |
| Cardiovascular disease | 7 (12.5) | 7 (38.9) | 0.0320 |
| Cerebrovascular disease | 2 (3.6) | 0 | NS |
| Others[e] | 14 (24.6) | 11 (61.1) | |
| No. of comorbidities listed above (%) | | | 0.0047 |
| None | 19 (33.4) | 1 (5.6) | |
| One | 16 (28) | 3 (16.7) | |
| Two | 9 (15.8) | 6 (33) | |
| Three or more | 13 (22.8) | 8 (44.4) | |
| Disease severity[f] (%) | | | 0.0218 |
| Moderate | 42 (73.7) | 8 (44.4) | |
| Severe | 15 (26.3) | 10 (55.6) | |

[a]IQR, interquartile range.
[b]NS, not significant.
[c]Calculated from the available data.
[d]COPD, chronic obstructive pulmonary disease.
[e]Gastrointestinal reflux, dyslipidemia, neoplasia, and dementia.
[f]Based on the masks.

Python module GSEApy was performed. The average degree and clustering coefficient were calculated for each community. The final network was built using Cytoscape, and biomarkers that were significantly associated with death were highlighted.

**Characteristics of the study population.** Among the 75 patients, 24% (18/75) succumbed to death in the hospital (Table 1). Patients were initially classified based on the type of ventilation received (nasal cannula [$n = 25$] or VentMask [$n = 25$] for moderate cases or CPAP [$n = 25$] for severe cases), and a significantly higher rate of fatal outcomes was observed for severe cases compared to moderate cases ($P = 0.0218$, chi-square test). Among the patients who died, comorbidities were nearly universally observed (94%; 17/18). Underlying conditions like hypertension were recorded for 46% (26/57) of the survivors and 50% (9/18) of the nonsurvivors, while cardiovascular diseases were significantly more common in the nonsurvivors ($P = 0.0320$, Fisher's exact test) compared to the survivors. Among the serological markers, lactate was significantly higher ($P < 0.0001$, Mann-Whitney test) in the nonsurvivors compared to the survivors, confirming its strong association to disease severity (12, 14).

**Metabolites associated with COVID-19-related in-hospital mortality.** In a logistic regression analysis (adjusted for age, gender, and BMI), 35 metabolites were significantly associated with COVID-19 mortality ($P < 0.05$), among which 10 biomarkers were significant at the unadjusted 0.025 level (Fig. 1A). Interestingly, cyclic AMP (cAMP) was significantly increased in the nonsurvivors compared to the survivors (odds ratio [OR], 7.4; 95% confidence interval [CI], 1.5 to 37). cAMP is a well-known intracellular messenger that functions as a regulator of various cellular activities, including cell growth and

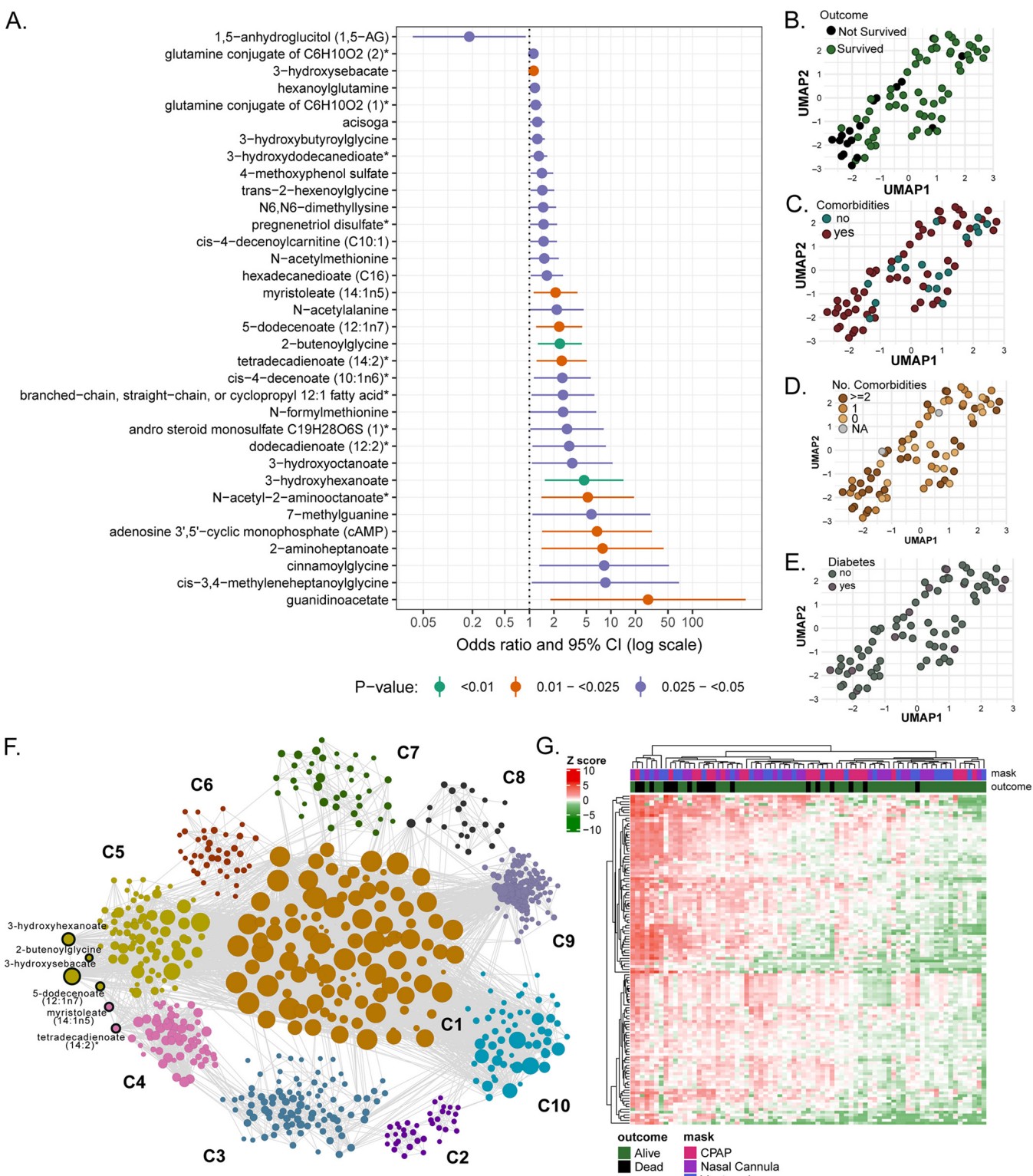

**FIG 1** Metabolites that are significantly associated with COVID-19-related in-hospital mortality. (A) All biomarkers that were significant at the 0.05 level, after adjusting for age, gender, and BMI, were included in the plot and ordered by the value of the odds ratio. Color coding for lower *P* values is presented in the legend. (B to E) UMAP visualization of patients' data after selecting the 10 death biomarkers. Patients were labeled for the outcome (B), presence of comorbidities (C), number of comorbidities (D), and presence of diabetes (E). (F) Global weighted network after community detection. Biomarkers with COVID-19-related mortality belong to communities 4 and 5 (labeled). The size of the bubble represents the connectivity of each metabolite. (G) Heat map of potential biomarkers and their first neighbors in patients using metabolite abundance levels from network analysis in patients labeled for mask types and outcome (survivors/nonsurvivors). The data were log-transformed, and hierarchical clustering was applied to both metabolites and patients.

differentiation, gene transcription, and protein expression (15), and it is intimately involved in mitochondrial dynamics (16). As cAMP plays a role in SARS-CoV-2 endocytosis in the initial phases of the infection (17), its involvement in the disease progression is worthy of further investigations as a potential biomarker.

**The identified metabolites do not correlate with patients' preexisting conditions or oxygen demand.** Using UMAP, we observed that the distribution of the patients in the biomarker enrichment showed a separation between the survivors and nonsurvivors (Fig. 1B). No specific pattern or clustering was observed for the presence of comorbidities (Fig. 1C), the number of comorbidities (Fig. 1D), or the presence of diabetes (Fig. 1E), indicating that these metabolite sets only differentiate related to the clinical outcome.

To further understand the patterns of metabolic changes related to COVID-19, we performed a weighted correlation network analysis. We identified 10 metabolite communities highly connected in a network of 916 nodes (metabolites) and 11,453 edges (Fig. 1F). Six predicted biomarkers (out of the 10 previously identified with $P < 0.025$, unadjusted) were highly correlated, were present in the network, and belonged to the lipid pathways (Fig. 1F). These metabolites are also known to be associated with peroxisomal fatty acid oxidation disorders (3-hydroxysebacate) (18) or insulin resistance [5-dodecenate (12:1n7), tetradecadienoate (14:2), and myristoleate (14:1n5)] (19).

Finally, to identify a larger cluster of metabolites associated with a fatal clinical outcome, we combined the results of the regression model with the network analysis. Inside the network, we selected the biomarkers identified from the regression models together with their first neighbors (metabolites correlated with these biomarkers). We represented this subset of metabolites (238 in total) in patients as a heat map, and we found a clear clustering of nonsurvivors in opposition to survivors (Fig. 1G). We show that patients with a fatal clinical outcome were present mostly in the same cluster. No clustering according to mask type was observed, indicating that the metabolic signature associated with mortality appears to be independent of the oxygen demand at the moment of hospitalization, providing the first identified correlation between a metabolic profile and disease severity in COVID-19 patients.

## CONCLUSIONS

Our analysis has identified metabolic biomarkers that differentiate between COVID-19 survivors and nonsurvivors and may be predictive of death from COVID-19, from the early stage of the epidemic, independently from oxygen demand at the moment of diagnosis. Our results on high-throughput metabolomics contribute to a better understanding of COVID-19-related metabolic disruption and may represent a useful starting point for the identification of independent prognostic factors to be employed in therapeutic practice. It is important to consider that when this study was planned (March to April 2020), there was little knowledge about COVID-19. Therefore, several clinical data were missing. Despite that, this is the first set of biomarkers identified from high-throughput metabolomics data that are associated with mortality and are not confounded by other preexisting conditions.

## ACKNOWLEDGMENTS

The study was funded by a Swedish Research Council grant (2017-01330) and Karolinska Institute Stiftelser och Fonder (2020-01554) to U.N. and by the Office of Research, University of Missouri, Columbia, USA, (F3593) to K.S. E.S. was partially supported by VR (2020-05836).

We thank all the patients involved in this study, as well as the dedicated medical and research staff fighting against SARS-CoV-2.

We acknowledge the work of the members of the COVID-19 Network Working Group:

Fondazione IRCCS Ca' Granda Ospedale Maggiore Policlinico, Milan, Italy

Scientific direction: Silvano Bosari (leader), Luigia Scudeller, Giuliana Fusetti, Laura Rusconi, Silvia Dell'Orto. Transfusion medicine (Biobank): Daniele Prati, Luca Valenti,

Silvia Giovannelli. Infectious Diseases Unit: Andrea Gori, Alessandra Bandera, Antonio Muscatello, Davide Mangioni, Laura Alagna, Giorgio Bozzi, Andrea Lombardi, Riccardo Ungaro, Teresa Itri, Valentina Ferroni, Valeria Pastore, Roberta Massafra, Ilaria Rondolini, Angelo Bianchi Bonomi. Internal Medicine, Hemophilia and Thrombosis Center and Fondazione Luigi Villa: Flora Peyvandi, Roberta Gualtierotti, Barbara Ferrari, Raffaella Rossio, Elisabetta Corona, Nicolò Rampi, Costanza Massimo. Internal Medicine, Immunology and Allergology: Nicola Montano, Barbara Vigone, Chiara Bellocchi, Giulia Coti, Mimma Sternativo. Respiratory Unit and Cystic Fibrosis Adult Center: Francesco Blasi, Stefano Aliberti, Maura Spotti, Edoardo Simonetta, Leonardo Terranova, Francesco Amati, Carmen Miele, Annalisa Vigni. Emergency Medicine: Giorgio Costantino, Monica Solbiati, Ludovico Furlan, Marta Mancarella, Giulia Colombo, Giorgio Colombo. Acute Internal Medicine: Valter Monzani, Angelo Rovellini, Filippo Billi, Christian Folli. Internal Medicine: Marina Baldini, Irena Motta. General Medicine and Metabolic Diseases: Anna Fracanzani, Rosa Lombardi. Geriatric Unit: Matteo Cesari, Marco Proietti. Istituto di Ricerche Farmacologiche Mario Negri IRCCS: Alessandro Nobili, Mauro Tettamanti.

U.N. and N.S.F. designed the study. E.S., A.B., M.S., F.M., A.A.L., S.A., M.C.S., F.B., F.B., E.E.G., G.C., P.D.R., S.K., A.G., F.P., and L.S. analyzed the data and performed the statistical analyses. E.S., U.N., and N.S.F. wrote the paper. U.N. and K.S. acquired the funding. E.S, C.L.L., L.V., K.S., L.B., N.S.F., and U.N. reviewed and edited the paper. All authors read and approved the final manuscript.

C.L.L. is the cofounder and CSO of Shift Pharmaceuticals. All other authors declare no competing conflicts of interest.

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
