## [Reviewer comments · Microbiology Spectrum]

Microbiology Spectrum

A distinct metabolic profile associated with a fatal outcome in COVID-19 patients during early epidemic in Italy

Elisa Saccon, Alessandra BANDERA, Mariarita Sciumè, Flora Mikaeloff, Abid Lashari, Stefano Aliberti, Michael Sachs, Filippo Billi, Francesco Blasi, Erin Gabriel, Giorgio Costantino, Pasquale Deroberto, Shuba Krishnan, Andrea Gori, Flora Peyvandi, Luigia Scudeller, Ciro Canetta, Christian Lorson, Luca Valenti, Kamalendra Singh, Luca Baldini, Nicola Fracchiolla, and Ujjwal Neogi

Corresponding Author(s): Ujjwal Neogi, Karolinska Institutet

Review Timeline:

Submission Date:	June 15, 2021
Editorial Decision:	July 13, 2021
Revision Received:	July 29, 2021
Accepted:	July 30, 2021

Editor: Tulip Jhaveri

Reviewer(s): Disclosure of reviewer identity is with reference to reviewer comments included in decision letter(s). The following individuals involved in review of your submission have agreed to reveal their identity: Nabil Benazi (Reviewer #1); Luciana Jesus Costa (Reviewer #4)

Transaction Report:

DOI: <https://doi.org/10.1128/Spectrum.00549-21>

July 13, 2021

Dr. Ujjwal Neogi
Karolinska Institutet
Laboratory Medicine
Alfred Nobel Allé 8, Plan 7
Huddinge, Stockholm 14186
Sweden

Re: Spectrum00549-21 (A distinct metabolic profile associated with a fatal outcome in COVID-19 patients during early epidemic in Italy)

Dear Dr. Ujjwal Neogi:

Thank you for submitting your manuscript to Microbiology Spectrum. When submitting the revised version of your paper, please provide (1) point-by-point responses to the issues raised by the reviewers as file type "Response to Reviewers," not in your cover letter, and (2) a PDF file that indicates the changes from the original submission (by highlighting or underlining the changes) as file type "Marked Up Manuscript - For Review Only". Please use this link to submit your revised manuscript - we strongly recommend that you submit your paper within the next 60 days or reach out to me. Detailed information on submitting your revised paper are below.

Link Not Available

Sincerely,

Tulip Jhaveri

Journals Department
Reviewer comments:

Reviewer #1 (Comments for the Author):

Despite the fact that this retrospective epidemiological analysis was only conducted for a short time and had a limited sample size, the topic is still relevant and important.

in the observation section :

Comment 1

The multivariate logistic regression must be detailed, that is, how you proceeded to the selection of these explanatory variables (univariate analysis: a simple logistic regression for each explanatory variable).

Comment 2

Because the statistical purpose of this study is to construct an explanatory final model, the initial model must appear next to the final model in table 1, together with the raw OR and final OR, their confidence intervals, and the P value (we propose have another table just for the regression).

Comment 3

The method for choosing the final model (stepwise regression method) must be mentioned in the paragraph (forward selection or backward selection, with a parsimony index such as the Akaike criterion (Akaike Information Criteria "AIC").

Comment 4

The "VIF" (Variance Inflation Factor) test for detecting collinearity between the selected variables must be specified in order to verify the lack of collinearity, which is a major problem in regressions.

Comment 5

A model fit test should also be mentioned, eg Hosmer - Lemeshow test.

In "Metabolites associated with COVID-19 related in-hospital mortality" section :

Comment 6

You mentioned that 35 metabolites were significantly associated with COVID-19 mortality ($p < 0.05$), but you only discussed one of them (cAMP) in your explanation, whereas there are other metabolites reported in your study that are important and deserve to be discussed to make the manuscript more relevant and robust.

Reviewer #2 (Comments for the Author):

Please brief up the BMI imputation using MICE and specifically mention the biomarkers considered in that case and also specify the markers considered for each case (e.g line no 135, 142, etc.).

Reviewer #3 (Comments for the Author):

The manuscript by Saccon et al presents data regarding the metabolic profile of 75 samples from COVID-19 patients with mild to severe disease. The data is relevant since it sets the path for a more in-depth investigation regarding the host response against SARS-CoV-2 infection that can predict the severity outcome of COVID-19.

Staff Comments:

Preparing Revision Guidelines

For complete guidelines on revision requirements, please see the Instructions to Authors at [link to page]. **Submissions of a paper that does not conform to Microbiology Spectrum guidelines will delay acceptance of your manuscript.**

Please return the manuscript within 60 days; if you cannot complete the modification within this time period, please contact me. If you do not wish to modify the manuscript and prefer to submit it to another journal, please notify me of your decision immediately so that the manuscript may be formally withdrawn from consideration by Microbiology Spectrum.

If you would like to submit an image for consideration as the Featured Image for an issue, please contact Spectrum staff.

Despite the fact that this retrospective epidemiological analysis was only conducted for a short time and had a limited sample size, the topic is still relevant and important.

in the observation section :

Comment 1

The multivariate logistic regression must be detailed, that is, how you proceeded to the selection of these explanatory variables (univariate analysis: a simple logistic regression for each explanatory variable).

Comment 2

Because the statistical purpose of this study is to construct an explanatory final model, the initial model must appear next to the final model in table 1, together with the raw OR and final OR, their confidence intervals, and the P value (we propose have another table just for the regression).

Comment 3

The method for choosing the final model (stepwise regression method) must be mentioned in the paragraph (forward selection or backward selection, with a parsimony index such as the Akaike criterion (Akaike Information Criteria "AIC").

Comment 4

The "VIF" (Variance Inflation Factor) test for detecting collinearity between the selected variables must be specified in order to verify the lack of collinearity, which is a major problem in regressions.

Comment 5

A model fit test should also be mentioned, eg Hosmer – Lemeshow test.

In “Metabolites associated with COVID-19 related in-hospital mortality” section :

Comment 6

You mentioned that 35 metabolites were significantly associated with COVID-19 mortality ($p < 0.05$), but you only discussed one of them (cAMP) in your explanation, whereas there are

other metabolites reported in your study that are important and deserve to be discussed to make the manuscript more relevant and robust.

Reviewer #1 (Comments for the Author):

Despite the fact that this retrospective epidemiological analysis was only conducted for a short time and had a limited sample size, the topic is still relevant and important.

Response: We appreciate the reviewer's positive comment regarding the impact of our study.

In the observation section

Comment 1

The multivariate logistic regression must be detailed, that is, how you proceeded to the selection of these explanatory variables (univariate analysis: a simple logistic regression for each explanatory variable).

Response: We are thankful to the reviewer for the comment. Using one biomarker at a time, we adjusted the analysis for a set of confounders that were identified based upon prior reports in the literature. No variable selection was done for confounders. We have modified the text to include an additional sentence that provides greater clarity (line 84):

"One biomarker was included in each model, but all other covariates were pre-specified for inclusion based on previous literature suggesting they are potential confounders."

Due to limited word count for observation, we mentioned the confounders in the figure legends

As follows "All biomarkers that were significant at the 0.05 level, after adjustment of age, gender, and BMI, were included in the plot and order by the value of the odds ratio."

Comment 2

Because the statistical purpose of this study is to construct an explanatory final model, the initial model must appear next to the final model in table 1, together with the raw OR and final OR, their confidence intervals, and the P value (we propose have another table just for the regression).

Response: we are thankful to the reviewer for the point. As stated above, there are no earlier models, as model selection was not used. The same pre-specified covariates were used in each model with a single biomarker. The only earlier models were those used to determine which biomarkers to include in the imputation. Please see response to reviewer #2 below for more detail. Within the text, we stated the goal of this work was "to identify a set of biomarkers strongly associated with the patient outcome". While an explanatory model is a potential outcome, we believe that identifying COVID-associated biomarkers is an important stand-alone outcome.

Comment 3

The method for choosing the final model (stepwise regression method) must be mentioned in the paragraph (forward selection or backward selection, with a parsimony index such as the Akaike criterion (Akaike Information Criteria "AIC")).

Response: As stated above there was no selection done for the "final model". We selected covariates based on previous literature indicating what may be confounding factors. We included a single biomarker in each regression with these covariates, in each model the same covariates were used.

Comment 4

The "VIF" (Variance Inflation Factor) test for detecting collinearity between the selected variables must be specified in order to verify the lack of collinearity, which is a major problem in regressions.

Response: We did not do variable selection, rather, we pre-specified the model based on previous literature and only include one biomarker with those covariates. Therefore, although it is possible that there is some collinearity in the pre-specified variables, we are not able to change it based on a test of collinearity without increasing the type one error rate. Pre-specification, in this case, is more important than any possible minor collinearity. As we are not doing any form of model selection and

each model contains the same three variables (age, BMI, and gender) in addition to one biomarker, this allows for a fair comparison of the biomarkers' association with the outcome. This provides an appropriate context to compare the relative importance of the individual biomarkers, regardless of any collinearity. Finally, there were no indications of non-convergence of the models, i.e. extremely large or extremely small SE, that would indicate near perfect collinearity. In the case of perfect collinearity, R automatically removes one of the variables.

Comment 5

A model fit test should also be mentioned, eg Hosmer - Lemeshow test.

Response: As we are not doing prediction based on the logistic model, the overall goodness of fit is not directly relevant to these analyses. We have a prespecified set of confounders (or potential confounders) that we include in each model. We are using these models to estimate associations with each of the biomarkers after adjustment for these known and pre-specified set of confounders, not for prediction of the outcome using the full model.

In "Metabolites associated with COVID-19 related in-hospital mortality" section:

Comment 6

You mentioned that 35 metabolites were significantly associated with COVID-19 mortality ($p < 0.05$), but you only discussed one of them (cAMP) in your explanation, whereas there are other metabolites reported in your study that are important and deserve to be discussed to make the manuscript more relevant and robust.

Response: We agree that the identified metabolites deserve a longer discussion for their known involvement in different cell functions and/or pathologies. Unfortunately, due to the required word limit (1200), we chose to focus on cAMP due to its possible involvement in SARS-CoV-2 infection and four others (3-hydroxysebacate, 5-dodecenate (12:1n7), tetradecadienoate (14:2)* and myristoleate (14:1n5)) for their strong association with the patients' outcome.

Reviewer #2 (Comments for the Author):

Please brief up the BMI imputation using MICE and specifically mention the biomarkers considered in that case and also specify the markers considered for each case (e.g line no 135, 142, etc.).

Response: We are thankful to the reviewer for the comment. We have added the following text to the paper (line no 86-91):

"As BMI was missing for many patients, multiple imputation by chained equations (MICE) was used (via the MICE R-package) to impute BMI using the predictive mean matching method. Imputation was done in two stages, first including no biomarkers, and then including those biomarkers that were found to be significant under the original imputation (see table for list of biomarkers). Fifty imputed datasets were created, and the resulting pooled estimates and inference were combined using Rubin's rules. P-values were not adjusted for multiple comparisons in this analysis."

Reviewer #4

The manuscript by Saccon et al presents data regarding the metabolic profile of 75 samples from COVID-19 patients with mild to severe disease. The data is relevant since it sets the path for a more in-depth investigation regarding the host response against SARS-CoV-2 infection that can predict the severity outcome of COVID-19.

Response: We are thankful to the reviewer for the positive comments on our manuscript.

Is the clustering of survivors versus non-survivors in figure 1G showing levels of metabolites? If that's the case, authors should explore better the results. The legend of figure 1 altogether should be more informative.

Response: Figure 1G shows the levels of the metabolites identified as death biomarkers in network analysis and their first neighbors.

We modified the text in the Observation:

“Finally, to identify a larger cluster of metabolites associated with a fatal clinical outcome, we combined the results of the regression model with the network analysis. Inside the network, we selected the biomarkers identified from the regression models together with their first neighbors (metabolites correlated with this biomarkers). We represented this subset of metabolites (238 in total) in patients as a heatmap and we found a clear clustering of non-survivors in opposition to survivors (Fig 1G). We show that patients with fatal clinical outcome were present mostly in the same cluster.”

We modified Figure 1G legend:

“Heatmap of potential biomarkers and their first neighbors in patients using metabolites abundance levels from network analysis in patients labeled for mask types and outcome (survivors/non-survivors). Data were log-transformed and hierarchical clustering applied on both metabolites and patients.”

On a minor note, the text needs a grammar/language revision.

Response: We have now thoroughly checked and modified the MS for the grammar/language errors.

July 30, 2021

Dr. Ujjwal Neogi
Karolinska Institutet
Laboratory Medicine
Alfred Nobel Allé 8, Plan 7
Huddinge, Stockholm 14186
Sweden

Re: Spectrum00549-21R1 (A distinct metabolic profile associated with a fatal outcome in COVID-19 patients during early epidemic in Italy)

Dear Dr. Ujjwal Neogi:

Your manuscript has been accepted, and I am forwarding it to the ASM Journals Department for publication. You will be notified when your proofs are ready to be viewed.

Sincerely,

Tulip Jhaveri
Editor, Microbiology Spectrum
